# A Novel Composite Hydrogel Material for Sodium Removal and Potassium Provision

**DOI:** 10.3390/polym15173568

**Published:** 2023-08-28

**Authors:** Jin Huang, Takehiko Gotoh, Satoshi Nakai, Akihiro Ueda

**Affiliations:** 1Graduate School of Advanced Science and Engineering, Hiroshima University, 1-4-1 Kagamiyama, Higashi-Hiroshima 739-8527, Hiroshima, Japan; d203238@hiroshima-u.ac.jp (J.H.); sn4247621@hiroshima-u.ac.jp (S.N.); 2Graduate School of Integrated Sciences for Life, Hiroshima University, 1-4-1 Kagamiyama, Higashi-Hiroshima 739-8527, Hiroshima, Japan; akiueda@hiroshima-u.ac.jp

**Keywords:** sodium ions, potassium ions, hydrogel, potassium fertilizer

## Abstract

Sodium ions are commonly found in natural water sources, and their high concentrations can potentially lead to adverse effects on both the water sources and soil quality. In this study, we successfully synthesized potassium polyacrylate (KMAA) hydrogel through free radical polymerization and evaluated its capability to remove sodium ions from and supply potassium ions to aqueous solutions. To assess its performance, inductively coupled plasma emission spectroscopy (ICP) was employed to analyze the sodium ion removal capacity and potassium ion exchange capability of the KMAA hydrogel at various initial sodium ion concentrations and pH values. The results demonstrated that the KMAA hydrogel exhibited remarkable efficiency in removing sodium ions and providing potassium ions. At pH 7, the maximum adsorption capacity for sodium ions was measured at 70.7 mg·g^−1^. The Langmuir model, with a correlation coefficient of 0.98, was found to be more suitable for describing the adsorption process of sodium ions. Moreover, at pH 4, the maximum exchange capacity for potassium ions reached 243.7 mg·g^−1^. The Freundlich model, with a correlation coefficient of 0.99, was deemed more appropriate for characterizing the ion exchange behavior of potassium ions. In conclusion, the successfully synthesized KMAA hydrogel demonstrates superior performance in removing sodium ions and supplying potassium ions, providing valuable insights for addressing high sodium ion concentrations in water sources and facilitating potassium fertilizer supply.

## 1. Introduction

Sodium ions, an essential nutrient for human health, can lead to various health problems such as hypertension and cardiovascular diseases when excessively consumed. The World Health Organization (WHO) has recommended a daily sodium intake not exceeding 2 g per day [1,2,3]. Nevertheless, natural water sources like rivers, lakes, and groundwater can contain elevated concentrations of sodium ions, due to the weathering of rocks and leaching from soils. This excessive sodium content can lead to soil salinization, posing a serious threat to crop growth. Consequently, it is imperative to develop efficient methods for removing sodium ions from aqueous solutions and mitigating soil salinization [4,5].

Currently, existing methods for sodium removal from water solutions, such as ion-exchange resin, reverse osmosis, distillation, electrodialysis, and vacuum evaporation, have limitations [6]. Some require high energy consumption, leading to environmental damage, while others are inefficient and less effective [7,8,9]. To overcome these challenges, hydrogels are promising, due to their unique properties as high-molecular-weight materials with a three-dimensional network structure. Hydrogels have a porous architecture and large surface area, making them efficient in adsorbing harmful substances in water and soil. Additionally, hydrogels can rapidly absorb significant amounts of water. Moreover, hydrogels are environmentally friendly, posing minimal pollution risk when applied in ecosystems [10,11].

Hydrogels, both natural and synthetic, offer promising solutions for sodium ion removal and soil salinization challenges. Recent advances in hydrogel research include superabsorbent hydrogels, salt-tolerant polymers, and innovative composites with improved soil water retention [12,13]. KMAA hydrogel adsorbent demonstrates exceptional water adsorption characteristics. It effectively captures trace sodium ions while concurrently releasing abundant potassium ions, thus serving as a valuable source of potassium fertilizer for plants with potassium requirements in the soil. Nonetheless, it is important to acknowledge that certain limitations exist within the methanol cleansing stage of the hydrogel preparation process. This becomes particularly relevant in the context of large-scale hydrogel synthesis, where a staged approach to cleansing might be warranted. Hydrogels also show positive effects on plant growth under drought and salt stress. Guo’s work showcased the successful synthesis of a cellulose-based superabsorbent hydrogel. This hydrogel exhibited extraordinary water uptake capacities of 604% in distilled water and 119% in salt water. Notably, the hydrogel demonstrated a remarkable ammonia nitrogen adsorption capacity of 30 mg·g^−1^, effectively mitigating nutrient leaching in soil [14]. Zhang’s contributions led to the development of a salt-tolerant superabsorbent polymer (ST-SAP) that exhibited exceptional swelling ability (69.04 g/g) under high salinity conditions [15]. Xiong’s pioneering work involved the synthesis of a novel superabsorbent polymer composite, wherein the addition of calcium alginate significantly enhanced soil water retention and facilitated controlled water release. Consequently, this composite exhibited increased water uptake rates of 25.8% and 10% in distilled water and salt water, respectively. The study highlights the immense potential of this approach in mitigating soil water evaporation and its application in saline–alkali soils in agriculture [16]. Tian’s research resulted in an exceptional superabsorbent polymer composite with outstanding water uptake abilities in various solutions, promising for water management and agriculture [17]. Shi’s investigation employed two hydrophilic polymers, Stockosorb and Luquasorb, to enhance the growth of one-year-old poplar cuttings under drought and salt stress conditions. Remarkably, the addition of 0.5% Stockosorb or Luquasorb significantly alleviated growth inhibition induced by drought and salt stress [18]. Islam’s study revealed that the combined application of salicylic acid (SA) and trehalose (Tre) had a more pronounced positive impact on mustard plants under sodium chloride (NaCl) stress compared to individual treatments [19]. Fathy’s pioneering work involved the synthesis of a stable composite cation exchange adsorbent through suspension polymerization, which exhibited an impressive 4.22 meq g^−1^ exchange capacity for Na^+^ ions. This research holds significant potential for further advancements in water purification and soil salinity management [20].

In this study, we synthesized KMAA hydrogel using free radical polymerization and explored its sustainable application strategies for sodium removal from water resources, sodium elimination in saline–alkali soils, and enhancement of potassium content as expressed in Figure 1. The primary aim was to investigate the sodium adsorption capacity and potassium exchange capacity of KMAA hydrogel under varying pH conditions, along with relevant thermodynamic and adsorption kinetic properties. To simulate the adsorption process, we employed the Langmuir and Freundlich isotherm models, while pseudo-first- and pseudo-second-order models were utilized to analyze the adsorption kinetics. Additionally, we assessed the impact of KMAA application in soil on sodium and potassium ion leaching, providing valuable insights for agricultural practices. Through these investigations, we aim to establish a theoretical foundation for effective sodium removal and potassium enrichment, offering feasible solutions for environmental restoration and agricultural production.

## 2. Experimental Methods

### 2.1. Materials

The monomer KMAA was obtained from Nippon Shokubai Co., Ltd. (Osaka, Japan). *N*, *N*′-dimethyl ethylenediamine (TEMED) was purchased from Nacalai Tesque Co., Ltd. (Kyoto, Japan). *N*, *N*′-methylenebisacrylamide (MBAA), ammonium persulfate (APS), and sodium chloride were acquired from Sigma-Aldrich, Inc. (St. Louis, MO, USA). Hydrochloric acid (HCl) was purchased from Sigma-Aldrich Japan (Tokyo, Japan). The soil was purchased from NAFCO (Fukuoka Japan), and its main elements were Si, Fe, Ca, Al, and Mn. The element contents detected by EDS were 49.1%, 18.8%, 10.2%, 9.6%, and 0.5%, respectively. All the reagents used were of analytical grade and employed as received. The distilled water used in the experiments was produced in the laboratory.

### 2.2. Synthesis of Hydrogel

A mass of 3.105 g of the monomer KMAA, 0.193 g of the cross-linking agent MBAA, and 0.058 g of the accelerator TEMED were weighed using an electronic balance and transferred into a 20 ml volumetric flask. Distilled water was added to the volumetric flask, and the mixture was stirred using a magnetic stirrer. Simultaneously, a mass of 0.057 g of the initiator APS was weighed and dissolved in 5 ml of distilled water to form an initiator solution. Both the monomer solution and the initiator solution were purged with nitrogen gas for 45 min to eliminate oxygen and prevent the inhibition of free radicals. Subsequently, the initiator solution was poured into the monomer solution while stirring for 25 s. The resulting mixture was transferred into three plastic tubes using a pipette. The monomer solution and the initiator solution were then immersed in a water solution at 25 °C for 24 h. The resulting gels were removed from the tubes and cut into uniform cylindrical shapes, and then washed with methanol for 24 h using Soxhlet extractor (Asahi Glass plant Inc., Arao City, Japan) equipment to remove any unreacted components. The gel was then air-dried at room temperature and further dried in an oven at 50 °C for 24 h. Finally, the gel was pulverized into powder using a grinder. The synthesis condition of the gel is summarized in Table 1.

### 2.3. Swelling Properties of Hydrogels

#### 2.3.1. Swelling Degree of Hydrogel in Sodium Chloride Solutions

A hydrogel weighing 0.02 g $\left(M_{d}\right)\;$ was added to 40 ml of sodium chloride solutions with concentrations of 10, 50, 100, 300, and 500 mg·L^−1^. The mixtures were allowed to equilibrate at room temperature for 24 h to achieve swelling equilibrium. The swollen gels were subsequently filtered through filter paper and their weight recorded as $M_{S},$ calculated by the following formula [21]:$$W_{a}=\frac{\left(Ms-Md\right)}{Md}\times 100\%$$

#### 2.3.2. Soil Water Holding Rate

Weighed soil of 10 g was combined with 0.1 g of hydrogel and recorded as $m_{1}$. Subsequently, an appropriate amount of water was added to the beaker at room temperature, allowing complete saturation of the mixture. The mass of the resulting mixture and water was measured by filtration using filter paper and recorded as $m_{2}$, calculated by the following formula:$$W_{b}=\frac{\left(m_{2}-m_{1}\right)}{m_{1}}\times 100\%$$

### 2.4. Adsorption Experiment

#### 2.4.1. Adsorption Thermodynamic Experiments with Different pH Values

The solutions were prepared by diluting 1000 ppm NaCl solution to concentrations of 10, 50, 100, 300, and 500 mg·L^−1^. Subsequently, 25 ml of each diluted solution was transferred to a plastic centrifuge tube with a capacity of 50 ml. The pH of the solutions was adjusted to 4.0 and 7.0 by adding 1 mol·L^−1^ of HCl solution to a final volume of 40 ml. Additionally, 20 mg of the adsorbent was added to each tube. The tubes were then placed in a water bath constant-temperature shaker at 25 °C with continuous shaking at 150 rpm for 24 h. After the adsorption period, the samples were filtered through a 0.22 μm membrane filter and analyzed using ICP.

#### 2.4.2. pH Effect Experiment

The 1000 ppm sodium chloride (NaCl) solution was diluted to a concentration of 100 mg·L^−1^. Subsequently, 25 ml of the diluted solution was transferred into a plastic centrifuge tube with a total volume of 50 ml. To adjust the pH values to 2.0, 3.0, 4.0, 5.0, 6.0, and 7.0, 1 mol·L^−1^ of HCl solution was added until the final volume reached 40 ml. Next, 20 mg of the adsorbent material was added to the mixture. The entire system was placed in a water bath constant-temperature shaker, where continuous adsorption was conducted at 25 °C and 150 rpm for a duration of 24 h. Following adsorption, the samples were filtered through a 0.22 μm membrane filter and analyzed using ICP.

#### 2.4.3. Adsorption Kinetic Experiment

The 1000 ppm NaCl solution was diluted to a concentration of 100 mg·L^−1^. Subsequently, 40 ml of the diluted solution was adjusted to pH 4.0 and 7.0 using 1 mol·L^−1^ of HCl solution. Next, 20 mg of the adsorbent material was introduced into the system. The entire setup was placed in a water bath constant-temperature shaker, maintaining a constant temperature of 25 °C with a shaking rate of 150 rpm. The adsorption process was conducted for specific time intervals (1, 10, 60, 180, 240, 300, 1200, 1380, and 1440 min). At each predetermined time point, samples were withdrawn from the system and subsequently filtered through a 0.22 μm membrane filter. The filtered samples were then analyzed using ICP measurement.

#### 2.4.4. Adsorption Thermodynamics at Different Temperatures

The 1000 ppm original NaCl solutions were diluted to concentrations of 10, 50, 100, 300, and 500 mg·L^−1^, respectively. For each concentration, 25 ml of the diluted solution was transferred into a plastic centrifuge tube with a volume of 50 ml. Subsequently, the pH of each solution was adjusted to 4.0 and 7.0 using 1 mol·L^−1^ of hydrochloric acid (HCl) solution to reach the final volume of 40 ml. Next, 20 mg of the adsorbent material was added to each solution. The prepared samples were placed in a water bath constant-temperature shaker, maintaining constant temperatures of 25 °C and 35 °C, respectively, with a shaking rate of 150 rpm. The adsorption process was conducted continuously for 24 h. After the adsorption period, the samples were filtered through 0.22 μm membrane filters to remove any particulate matter. The filtrates were subsequently analyzed using ICP to measure the desired parameters.

#### 2.4.5. Adsorbent Dosage Experiment

The 1000 ppm NaCl solution was diluted to a concentration of 100 mg·L^−1^. Subsequently, 25 ml of the diluted solution was transferred into a plastic centrifuge tube with a volume of 50 ml. The pH of each solution was adjusted to 4.0 and 7.0 by adding 1 mol·L^−1^ of HCl solution until the final volume reached 40 ml. Next, different amounts of the adsorbent 3, 5, 10, 15, and 20 mg were added to the samples. The prepared samples were subjected to continuous adsorption for 24 h in a water bath constant temperature shaker at 25 °C with a shaking rate of 150 rpm. After the adsorption period, the samples were filtered through a 0.22 μm membrane filter to remove any particulate matter. The filtrates were then analyzed using ICP.

### 2.5. Soil Experiment

#### 2.5.1. Preparation of Soil Containing NaCl

Three separate beakers, labeled A1, A2, and A3, were used for the soil sample analysis. An amount of 10 g of soil was weighed and placed into each respective beaker. Beaker A1 served as the blank control, while beaker A2 received the addition of 5 ml of a 1000 ppm NaCl solution. Similarly, beaker A3 also received 5 ml of a 1000 ppm NaCl solution, along with an added 0.2 g of KMAA hydrogel. To ensure proper mixing, an appropriate amount of water was added to each beaker, and thorough stirring was conducted. Subsequently, the beakers were transferred to an oven for drying.

#### 2.5.2. pH Value Affects the Precipitation of Sodium and Potassium Ions in Soil

Two aqueous solutions with pH values of 4 and 7 were prepared by measuring out 40 ml of each solution. Subsequently, 4 g soil samples (A1, A2, and A3) were individually added to the separate beakers, followed by the addition of 40 ml of the respective pH solution. The solid-to-liquid ratios were maintained at 1:10. The soil solutions were shaken for 24 h in a water bath constant-temperature shaker at 25 °C with a shaking rate of 150 rpm. Afterward, the mixtures were filtered through a 0.22 μm membrane filter and analyzed using ICP.

#### 2.5.3. Amount of Hydrogel Affects Precipitation of Sodium and Potassium Ions in Soil

Two aqueous solutions were prepared by combining 40 ml of pH 4 and pH 7 solutions. Subsequently, 4 g soil samples containing 2% and 4% hydrogels were individually added to separate containers. The mixtures were then placed in a water bath constant-temperature shaker at 25 °C with continuous shaking at 150 rpm for 24 h. Following this, the mixtures were filtered using a 0.22 μm membrane filter and subsequently analyzed by ICP.

## 3. Result and Discussion

### 3.1. Swelling Degree of Hydrogel

This study investigates the water retention capacity of a hydrogel prepared for soil application. Upon incorporation of the hydrogel into the soil, a remarkable enhancement in its water-holding capability was observed, as depicted in Figure 2a. The soil’s water retention capacities were measured to be 187% without the hydrogel and 452% with the hydrogel. However, after 6 days in the presence of the hydrogel, the soil’s water-holding capacity diminished to zero. The superior water retention ability of the soil can be attributed to the hydrogel’s three-dimensional network structure and surface hydrophilic groups, which facilitate the diffusion of water molecules into the hydrogel’s internal matrix. Furthermore, the hydrophilic groups within the polymeric network of the hydrogel underwent ionization with the internal solution, leading to the generation of osmotic pressure and continuous diffusion of water molecules into the interconnected network of the hydrogel until reaching the equilibrium swelling state. Consequently, the introduction of the hydrogel significantly strengthened the soil’s water retention capacity. In Figure 2b, it can be observed that the swelling capacity of the hydrogel in the sodium chloride solution decreased with increasing sodium chloride concentration. At a 500 ppm sodium chloride concentration, the hydrogel exhibited the lowest swelling capacity, measuring 135%. This decline in swelling can be attributed to the formation of osmotic pressure, both inside and outside the hydrogel, thereby suppressing the inward diffusion of water. At high concentrations of sodium chloride, the hydrogel’s structure could even be compromised, resulting in a reduction of its water absorption performance. In summary, this study reveals the potential of hydrogels in improving soil water retention, thereby providing valuable insights for future applications in agriculture and water management [22].

### 3.2. pH Affects the Adsorption of the Gel

This study investigates the performance of KMAA hydrogel in the removal of Na^+^ and the exchange of K^+^ under different pH conditions, as depicted in Figure 3a. The removal efficiency of Na^+^ decreases with increasing sodium chloride concentration when the same KMAA hydrogel is added. At a 10 ppm sodium chloride concentration, the highest removal efficiency for Na^+^ reaches 62%. Conversely, the exchange rate of K^+^ increases with higher concentrations of sodium chloride, with the maximum exchange rate of 48% observed at a 500 ppm sodium chloride concentration. Figure 3b illustrates the adsorption of Na^+^ and the exchange of K^+^ by the KMAA hydrogel under various pH conditions in a 100 ppm sodium chloride solution. The adsorption of sodium ions by the KMAA hydrogel increases with the higher pH values. However, at pH 2 and 3, the hydrogel exhibits no adsorption of Na^+^. This phenomenon can be attributed to the competition between the highly concentrated hydrogen ions (H^+^) in the strong acid condition and the carboxylic acid groups on the hydrogel, leading to a further increase in the protonation degree of carboxylic acid. Consequently, the quantity of negatively charged carboxylate ions on the hydrogel surface is fundamentally reduced, resulting in a diminished binding capacity with the sodium ions in the solution. Therefore, under strong acidic conditions, the KMAA hydrogel exhibits minimal capacity to adsorb Na^+^. On the other hand, the exchange capacity of K^+^ by the KMAA hydrogel decreases with increasing pH of the solution. At pH = 2, the hydrogel displays the highest K^+^ exchange capacity, reaching 249 mg·g^−1^. As the solution’s acidity increases, the concentration of hydrogen ions also rises. These hydrogen ions (H^+^) form associations with carboxyl groups (COO^−^) and create a repulsive force with the potassium ions, enhancing the exchange capacity of K^+^ on the hydrogel surface.

### 3.3. Isothermal Adsorption

The adsorption of sodium ions and the exchange of potassium ions by the KMAA hydrogel at pH = 4 and pH = 7 were investigated, and the isotherms are depicted in Figure 4. To analyze the adsorption behavior, two isothermal adsorption models, Langmuir and Freundlich, are used for fitting.
$$q_{e}=\frac{q_{max}K_{L}C_{e}}{1+K_{L}C_{e}}$$
$$q_{e}=K_{F}C_{e}^{1/n}$$

In the equations, $q_{e}$ represents the adsorption capacity at equilibrium (mg·g^−1^), $q_{max}$ is the maximum adsorption capacity calculated from the Langmuir model (mg·g^−1^), $K_{L}$ is the Langmuir constant (L·mg^−1^), and 1/n and $K_{F}$ are the constants of the Freundlich model (mg·g^−1^). With increasing sodium chloride concentration, the adsorption of sodium ions and the exchange of potassium ions by KMAA hydrogel both increase. The Langmuir model assumes that the adsorption of the adsorbent to the target is only single-layer adsorption, and the interaction force between the adsorbed molecules can be ignored. The Freundlich model assumes that the adsorption of the adsorbent to the target is heterogeneous multilayer adsorption. According to the Langmuir model calculation, the maximum adsorption capacity of KMAA hydrogel for sodium ions at pH = 7 reaches 80.5 mg·g^−1^, while the actual maximum adsorption capacity is 70.7 mg·g^−1^. From the Freundlich model calculation, the maximum exchange capacity of potassium ions at pH = 4 reaches 254.6 mg·g^−1^, and the actual maximum exchange capacity is 243.7 mg·g^−1^. The KMAA hydrogel exhibits a three-dimensional network structure with high selective adsorption properties. The value of parameter n in the Freundlich model ranges from 1 to 10, indicating a spontaneous adsorption process.

Table 2 shows that the correlation coefficients of the Langmuir model are higher than those of the Freundlich model at different pH values. At pH = 4 and 7, the correlation coefficients of the Langmuir model are 0.98 and 0.97, respectively, indicating that the Langmuir model is more suitable for describing the adsorption process of sodium ions. Table 3 reveals that the correlation coefficients of the Freundlich model are higher than those of the Langmuir model. At pH = 4 and 7, the correlation coefficients of the Freundlich model are 0.9934 and 0.9936, respectively, suggesting that the Freundlich model is more appropriate for describing the exchange of potassium ions.

### 3.4. Adsorption Thermodynamics

To investigate the thermodynamic properties of the adsorption of sodium ions and the exchange of potassium ions by KMAA hydrogel at pH = 4and 7, the calculations of Gibbs free energy (∆*G*), enthalpy change (∆*H*), and entropy change (∆*S*) were performed. The formulas for these calculations are as follows [23]:$$K_{D}=\frac{q_{e}}{C_{e}}$$
$$∆G^{\theta }=-RTInK_{D}$$
$$∆G^{\theta }=∆H^{\theta }-T∆S^{\theta }$$
In the equations, $K_{D}$ represents the distribution coefficient, R is the ideal gas constant (8.314 J·mol^−1^·K^−1^), *T* is the thermodynamic temperature (K), $∆G^{\theta }$ represents the Gibbs free energy (kJ·mol^−1^), $∆H^{\theta }$ represents the enthalpy change (kJ·mol^−1^), and $∆S^{\theta }$ represents the entropy change (J·mol^−1^·K^−1^). By calculating the thermodynamic parameters, including the adsorption enthalpy ($∆H^{\theta }$), adsorption free energy ($∆G^{\theta }$), and adsorption entropy ($∆S^{\theta }$), the adsorption mechanism of the hydrogel was studied. A thermodynamic analysis was performed to elucidate the effect of different temperatures on the adsorption of sodium ions and the exchange of potassium ions on the KMAA hydrogel, as shown in Figure 5. Table 4 and Table 5 show the KMAA hydrogel adsorption thermodynamic parameters for sodium and for potassium, respectively. It was found that when the temperature increased from 298 K to 318 K, $∆G^{\theta }$ remained negative, indicating that the adsorption of sodium ions and the exchange of potassium ions on KMAA hydrogel are spontaneous and feasible processes. For the adsorption of sodium ions, the $∆H^{\theta }$ was positive, and the adsorption capacity of sodium ions increased with temperature, indicating an endothermic reaction. Additionally, the $∆S^{\theta }$ was positive, indicating an increase in disorder at the interface between the hydrogel surface and sodium ions during the adsorption process. Regarding the exchange of potassium ions on the hydrogel, as the temperature increased from 298 K to 318 K, the $∆H^{\theta }$ was negative, suggesting that the exchange process is exothermic. Simultaneously, the $∆S^{\theta }$ was also negative, indicating a decrease in randomness at the solid/liquid interface during the exchange process. Thermodynamic investigations provide insights into whether the adsorption process is endothermic or exothermic, thereby expanding the practical application of adsorbents. Overall, the thermodynamic study enables a better understanding of the adsorption behavior of the hydrogel, including whether it is exothermic or endothermic, and contributes to the broader application of adsorbents.

### 3.5. Adsorption Kinetics

The study of adsorption kinetics can reflect the rate and mechanism of the adsorption target of the adsorbent material. The exchange kinetics of the KMAA hydrogel for the adsorption of sodium and potassium are shown in Figure 6. Two kinds of isotherm adsorption models, the pseudo-first-order kinetic model and the pseudo-second-order kinetic model, were used for fitting [24].
$$q_{t}=q_{e}(1-e^{-k_{1}t})$$
$$q_{t}=\frac{\left(k_{2}q_{e}^{2}t\right)}{1+k_{2}q_{e}t}$$

In the equations, $q_{t}$ represents the adsorption capacity at time t (mg·g^−1^), $q_{e}$ represents the adsorption capacity at equilibrium (mg·g^−1^), $k_{1}$ is the rate constant of the pseudo-first-order kinetics (min^−1^), and $k_{2}$ is the rate constant of the pseudo-second-order kinetics (g·mg^−1^·min^−1^). The entire adsorption process can be divided into two stages. In the initial stage, the adsorption rate is rapid, and the adsorption capacity of sodium ions and the exchange of potassium ions by the KMAA hydrogel reaches approximately 90% of the equilibrium adsorption capacity after 10 min. As time progresses, the rate of sodium ion adsorption and potassium ion exchange gradually decreases, and equilibrium is nearly reached after 24 h. This may be attributed to the dissociation of -COOK groups on the hydrogel surface, leading to the formation of many COO^−^ and K^+^ ions in the solution during the initial stage. The -COO^−^ sites bind extensively with sodium ions. With the increase in time, many Na^+^ ions bind to -COO^−^ sites, resulting in an equilibrium in the osmotic pressure of the solution and the hydrogel. This leads to the adsorption of sodium ions, a decrease in the exchange of potassium ions, and, finally, the adsorption equilibrium is reached.

From Table 6, it can be observed that, at pH = 4 and 7, the coefficients of the pseudo-second-order kinetic model for sodium ion adsorption are consistently higher than those of the pseudo-first-order kinetic model, indicating the presence of chemical adsorption during the sodium adsorption. Thus, the pseudo-second-order kinetic model is more suitable for describing the adsorption process of sodium ions. From Table 7, it is evident that, at pH = 4 and 7, the coefficients of the pseudo-second-order kinetic model for potassium ion exchange are lower than those of the pseudo-first-order kinetic model. This suggests that the exchange process of potassium ions is mainly governed by electrostatic repulsion, indicating that the pseudo-first-order kinetic model is more appropriate for describing the exchange process of potassium ions. 

### 3.6. The Effect of Hydrogel Dosage

In practical applications of adsorption materials, the quantity of added material is a critical parameter. As depicted in Figure 7, the initial sodium chloride concentration was 100 mg·L^−1^, and the amount of KMAA hydrogel added varied from 3 mg to 20 mg. The results reveal that with an increase in the amount of KMAA hydrogel, the removal efficiency of Na^+^ rose from 13% to 32%, while the exchange capacity for K^+^ decreased from 37% to 26%. Notably, with an addition of 3 mg of hydrogel, the adsorption capacity for Na^+^ and K^+^ exchange reached their lowest values of 20 mg·g^−1^ and 25 mg·g^−1^, respectively. Conversely, with a higher addition of 20 mg, the KMAA hydrogel exhibited its maximum adsorption capacity for Na^+^ at 60 mg·g^−1^ and K^+^ exchange at 130 mg·g^−1^. These findings underscore a significant enhancement in adsorption and exchange performance with the increasing quantity of the KMAA hydrogel. These results hold paramount importance for guiding the practical application of adsorption materials.

### 3.7. Hydrogel Applied Soil Experiments

#### 3.7.1. Soil Precipitation Liquid Experiment

The results of the experiments studying the addition of KMAA hydrogel to sodium-containing soils are shown in Figure 8. The concentration of sodium in the soil filtrate decreased with the addition of the hydrogel. In the control group without hydrogel and the group with added sodium, the concentration of sodium in the soil filtrate was 120 ppm and 116 ppm at pH = 4 and pH 7, respectively. When 2% hydrogel was added, the concentration of sodium in the soil filtrate decreased to 88 ppm and 85 ppm, representing a reduction of 26%. Conversely, the concentration of potassium in the filtrate increased with the addition of the hydrogel. In the control group without hydrogel and the group with added sodium, the concentration of potassium in the soil filtrate was 184 ppm and 180 ppm at pH = 4 and pH 7, respectively. When 2% hydrogel was added to the soil, the concentration of potassium in the filtrate increased to 317 ppm and 309 ppm, showing a 72% increase. These results indicate that the addition of KMAA hydrogel effectively reduces the concentration of sodium in the soil filtrate while increasing the concentration of potassium. This suggests that the hydrogel acts as a selective adsorbent, selectively adsorbing sodium ions and promoting the release of potassium ions into the filtrate. This finding has significant implications for soil remediation and nutrient management in agricultural practices.

#### 3.7.2. Effect of Different Gel Amounts on Precipitate

Adding different amounts of gel to the soil and recording the concentration of sodium and potassium in the filtrate was also studied. From Figure 9, it can be observed that, with an increase in the amount of hydrogel, the concentration of sodium in the soil filtrate did not decrease at pH = 4 and pH = 7. This suggests that sodium may have been replaced by other ions present in the soil. However, at pH = 4 and pH = 7, the concentration of potassium in the soil filtrate increased with the addition of the hydrogel. Sodium is known to be a harmful element for plant growth, while potassium is an essential nutrient for plant growth. By examining the sodium and potassium content in the filtrate, it is possible to determine the appropriate amount of hydrogel to add. This approach can help mitigate the harmful effects of sodium on plants while providing an adequate supply of potassium for plant growth.

### 3.8. Structural Characteristics of Hydrogel

The chemical structure of the KMAA was detected by Fourier-transform infrared spectroscopy (FTIR). Figure 10 shows the infrared images of the KMAA hydrogel before and after adsorption in a 100 ppm sodium chloride solution. The pristine KMAA hydrogel exhibited distinct absorption peaks: the C-C bond displayed a peak at 1128 cm^−1^, the C=C double bond featured at 1517 cm^−1^, the C=O moiety of the carboxyl group manifested at 1720 cm^−1^, and the O-H band of the carboxyl group was evident at 3579 cm^−1^. After adding the KMAA hydrogel to a 100 ppm sodium chloride solution for adsorption, the ensuing alterations were observed: the C-C single bond peak shifted to 1186 cm^−1^, and the C=C double bond peak shifted to 1537 cm^−1^. The C=O peak of the carboxyl group exhibited a marginal redshift to 1639 cm^−1^, and the C-H peak of the methyl group emerged at 2927 cm^−1^. The O-H peak of the carboxyl group exhibited a redshift to 3269 cm^−1^ [25]. These changes collectively suggest the adsorption of sodium ions from the solution by the hydrogel, coupled with the concomitant release of potassium species. These structural transformations can be attributed to the adsorptive interaction of the hydrogel in the sodium chloride solution, leading to an augmented intermolecular irregularity. Consequently, an enhanced hydrogen bonding effect between the solution and the hydrogel is invoked, thereby inducing the redshift phenomena observed in the O-H and C=O bands.

### 3.9. Mechanism of KMAA Hydrogel Adsorbing Sodium and Exchanging Potassium

Potassium methacrylate polymer is a kind of high water-absorbing polymer, which will dissociate into carboxylate ions and potassium ions in solution, and the pH value will affect the dissociation degree of the hydrogel. Potassium ions and sodium ions belong to the same group of elements, and their atomic radii are very similar. In the solution, through the form of cation exchange, sodium ions are adsorbed by the hydrogel, and potassium ions will be replaced in the aqueous solution. In addition, the stronger the acidity, the more H^+^ value in the solution, which is more conducive to the release of potassium ions into the solution, but, at the same time, the amount of sodium adsorption is reduced [26]. The novelty of the experiment is an in-depth study of the experimental evaluation of the Na ion adsorption and potassium ion release of the KMAA hydrogel in acidic and neutral solutions, providing tangible evidence to bolster their potential utility in soil applications. Looking forward, KMAA hydrogels exhibit promise as a plausible potassium fertilizer for soil application, addressing the crucial need for essential potassium nutrients in plants. This application holds promise for various crops, encompassing maize cultivation, among others. Moreover, these hydrogels display the capability to mitigate mildly saline–alkaline soils to a certain extent, making them suitable for cultivating crops like rice. These insights open fresh avenues and possibilities for innovative soil amendments within the realm of agricultural practices.

## 4. Conclusions

In this study, we successfully synthesized potassium polyacrylic acid (KMAA) hydrogel using free radical polymerization and investigated its pH-sensitive properties. The results demonstrated that the hydrogel exhibited remarkable efficiency in sodium ion adsorption under neutral conditions (pH = 7), while it demonstrated significant potassium ion release under acidic conditions (pH = 4). To gain a deeper understanding of the adsorption process, we employed the Langmuir and Freundlich isotherm models, as well as pseudo-first-order and pseudo-second-order kinetic models, to conduct a comprehensive analysis. For the pH = 7 conditions, the Langmuir model provided a good fit to the experimental data with a correlation coefficient of 0.98, indicating its suitability for describing the adsorption of sodium ions. On the other hand, for the pH = 4 conditions, the Freundlich model exhibited a better fit with a correlation coefficient of 0.99, suggesting its appropriateness for describing the exchange of potassium ions. Thermodynamic analysis revealed that the standard free energy change (*ΔG*^θ^) for sodium adsorption and potassium exchange by the KMAA hydrogel was negative at both pH = 4 and pH = 7, indicating the spontaneity and feasibility of these processes. Upon introducing 2% hydrogel into the soil, we observed a 26% reduction in sodium concentration and a 72% increase in potassium concentration in the filtrate [27,28].

## Figures and Tables

**Figure 1 polymers-15-03568-f001:**
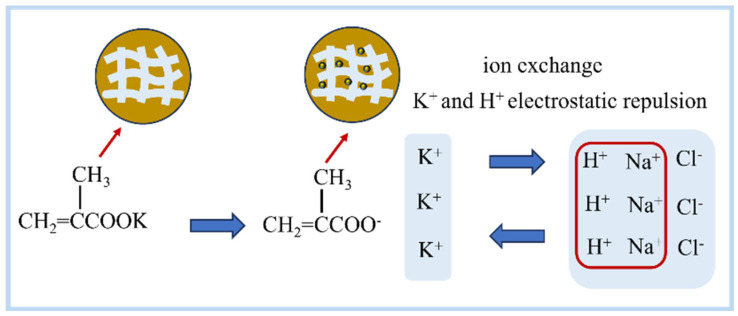
Schematic diagram of the mechanism of KMAA hydrogel to adsorb sodium and provide potassium in water and soil.

**Figure 2 polymers-15-03568-f002:**
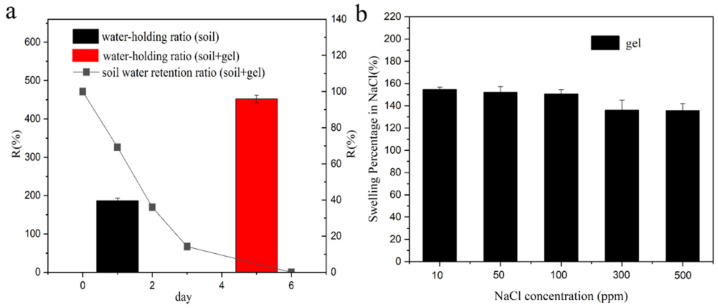
(**a**) Water retention capability of hydrogel in soil, (**b**) swelling behavior of hydrogel in NaCl solution.

**Figure 3 polymers-15-03568-f003:**
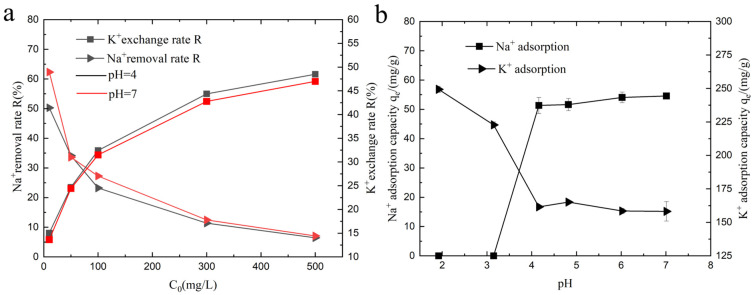
(**a**) Na^+^ removal efficiency and K^+^ exchange rate of hydrogel under various initial NaCl concentrations. (**b**) Effect of different pH values on Na^+^ adsorption and K^+^ exchange capacity of hydrogel.

**Figure 4 polymers-15-03568-f004:**
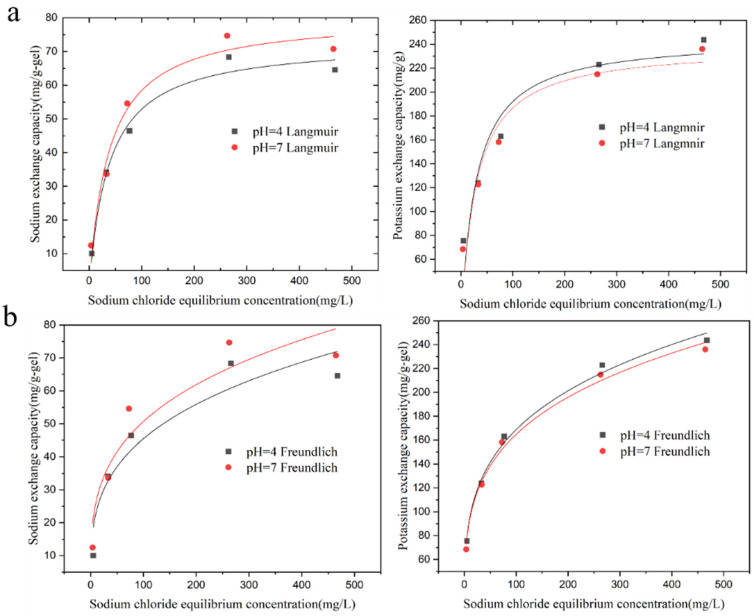
Isothermal adsorption of Na^+^ adsorption and K^+^ exchange by hydrogel at pH = 4 and 7. (**a**) Langmuir model simulated fit curve, (**b**) Freundlich model simulated fit curve.

**Figure 5 polymers-15-03568-f005:**
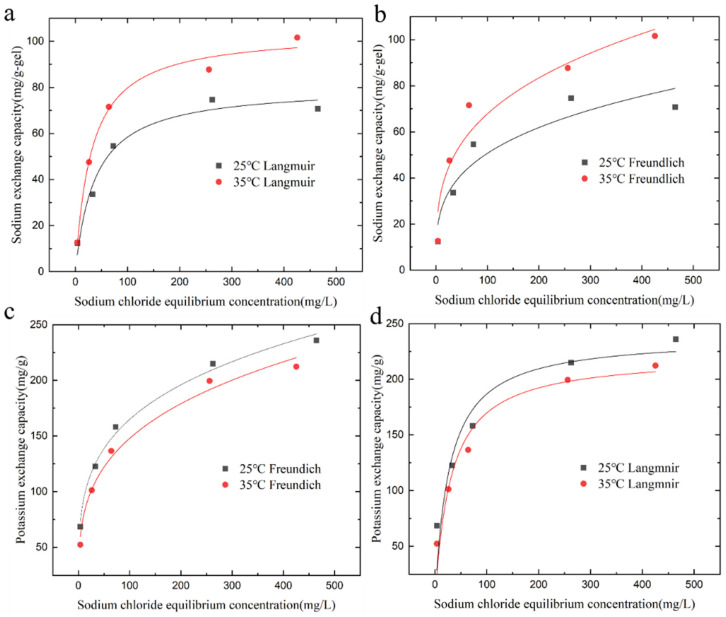
Adsorption isotherm fit curves of KMAA hydrogel at different temperatures. (**a**) Na^+^ adsorption Langmuir model simulated fit curve, (**b**) Na^+^ adsorption Freundlich model simulated fit curve, (**c**) K^+^ exchange Langmuir model simulated fit curve, and (**d**) K^+^ exchange Freundlich model simulated fit curve.

**Figure 6 polymers-15-03568-f006:**
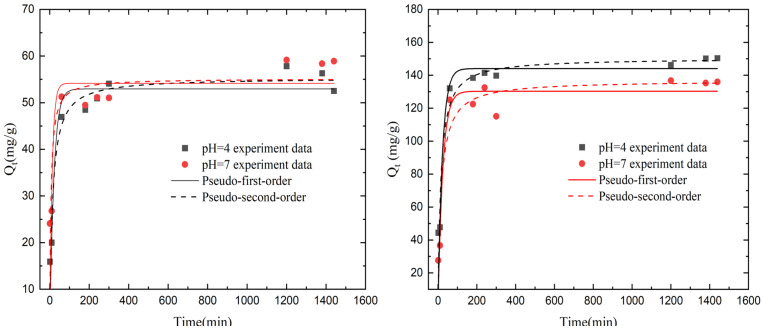
(**Left**) Dynamic modeling of pseudo-first-order and pseudo-second-order Na^+^ adsorption by KMAA hydrogel at initial NaCl concentration of 100 mg·L^−1^; (**Right**) dynamic modeling of pseudo-first-order and pseudo-second-order K^+^ exchange by KMAA hydrogel at initial NaCl concentration of 100 mg·L^−1^.

**Figure 7 polymers-15-03568-f007:**
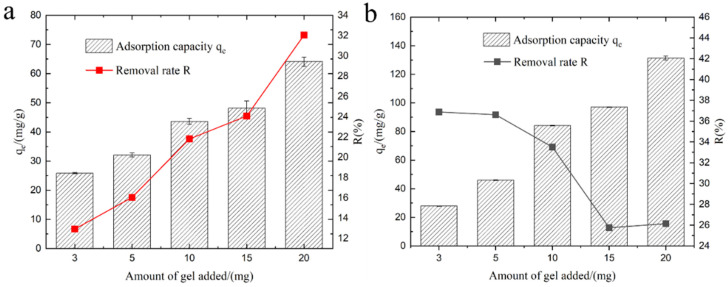
(**a**) Effect of different dosages of KMAA hydrogel on Na^+^ adsorption capacity and sodium removal efficiency. (**b**) Effect of different dosages of KMAA hydrogel on K^+^ exchange capacity and potassium exchange rate.

**Figure 8 polymers-15-03568-f008:**
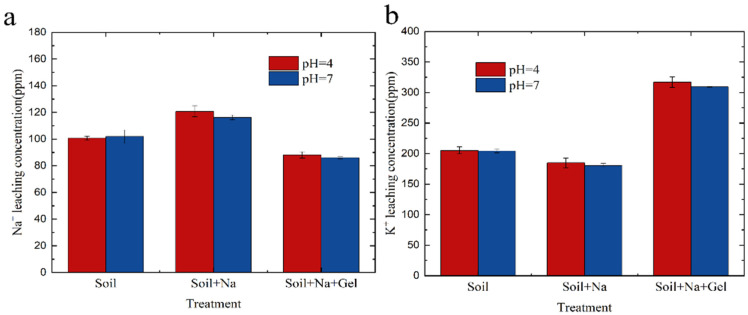
(**a**) Na^+^ leaching in soil solutions at different pH levels. (**b**) K^+^ leaching in soil solutions at different pH levels.

**Figure 9 polymers-15-03568-f009:**
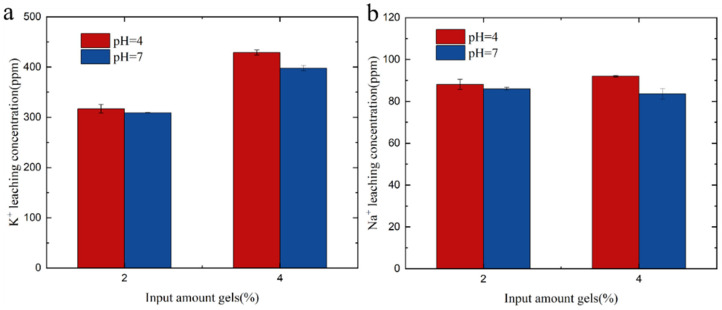
(**a**) K^+^ leaching in soil solutions with varying gel amounts. (**b**) Na^+^ leaching in soil solutions with varying gel amounts.

**Figure 10 polymers-15-03568-f010:**
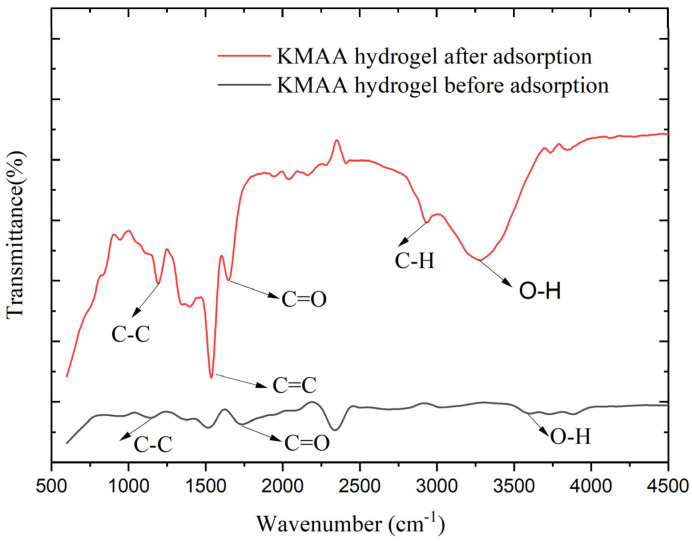
FTIR spectrum of gels before and after adsorption at 600~4500 (cm^−1^).

**Table 1 polymers-15-03568-t001:** Synthesis conditions of the KMAA hydrogel.

Materials	Component Type	Molar Weight (g·mol^−1^)	Concentration (mol·m^−3^)	Mass (g)
KMAA	Monomer	124.18	1000	3.105
MBAA	Linker	154.17	50	0.193
TEMED	Accelerator	116.21	20	0.058
APS	Initiator	228.19	10	0.057

**Table 2 polymers-15-03568-t002:** Parameters of Langmuir and Freundlich isotherm models of KMAA gel at pH = 4 and pH = 7 sodium ion adsorption capacity.

Isotherm Model	Parameter	Initial pH	
		4	7
Langmuir	*q*_max_ (mg·g^−1^)	72.9	80.5
	*K_L_* (L·mg^−1^)	0.0264	0.0265
	*R^2^*	0.98	0.97
Freundlich	*K_F_* (mg·g^−1^)	11.54	13.47
	*n*	3.448	3.571
	*R* ^2^	0.909	0.905

**Table 3 polymers-15-03568-t003:** Parameters of Langmuir and Freundlich isotherm models of KMAA gel at pH = 4 and pH = 7 potassium ion exchange capacity.

Isotherm Model	Parameter	Initial pH	
		4	7
Langmuir	*q*_max_ (mg·g^−1^)	254.6	238.4
	*K_L_* (L·mg^−1^)	0.0358	0.0361
	*R^2^*	0.8964	0.8931
Freundlich	*K_F_* (mg·g^−1^)	52.37	52.26
	*n*	3.937	4.016
	*R* ^2^	0.9934	0.9936

**Table 4 polymers-15-03568-t004:** KMAA hydrogel adsorption thermodynamic parameters for sodium.

T/K	${∆G}^{\theta }/($kJ·mol^−1^)	${∆H}^{\theta }/($kJ·mol^−1^)	${∆S}^{\theta }/($J·mol^−1^·K^−1^)
298.15	−2.965	2.762	19.21
308.15	−3.157	2.762	19.21

**Table 5 polymers-15-03568-t005:** KMAA hydrogel exchange thermodynamic parameters for potassium.

T/K	${∆G}^{\theta }/($kJ·mol^−1^)	${∆H}^{\theta }/($kJ·mol^−1^)	${∆S}^{\theta }/($J·mol^−1^·K^−1^)
298.15	−1.406	−1.589	−44.21
308.15	−0.964	−1.589	−44.21

**Table 6 polymers-15-03568-t006:** Fit results of Na^+^ adsorption kinetics by KMAA hydrogel.

Isotherm Model	Parameter	Initial pH	
		4	7
Pseudo-first-order	*k*_1_ (min^−1^)	0.049	0.084
	*q*_e_ (mg·g^−1^)	52.94	54.13
	*R^2^*	0.872	0.631
Pseudo-second-order	*k*_2_ (g·mg^−1^·min^−1^)	0.00133	0.0312
	*q*_e_ (mg·g^−1^)	55.34	57.31
	*R* ^2^	0.90191	0.714

**Table 7 polymers-15-03568-t007:** Fit results of K^+^ adsorption kinetics by KMAA hydrogel.

Isotherm Model	Parameter	Initial pH	
		4	7
Pseudo-first-order	*k*_1_ (min^−1^)	0.00044	0.04131
	*q*_e_ (mg·g^−1^)	136.82	130.28
	*R^2^*	0.92195	0.935
Pseudo-second-order	*k*_2_ (g·mg^−1^·min^−1^)	0.000467	0.04468
	*q*_e_ (mg·g^−1^)	150.41	144.08
	*R* ^2^	0.89644	0.89122

## Data Availability

Not applicable.

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
