# Peer review of "A Novel Composite Hydrogel Material for Sodium Removal and Potassium Provision"

_polymers, 2023, doi:10.3390/polym15173568_

Round 1

Reviewer 1 Report

In this study, the authors successfully synthesized potassium polyacrylate hydrogel through free radical polymerization and evaluated its capability to remove sodium ions and supply potassium ions from aqueous solutions. To assess its performance, inductively coupled plasma emission spectroscopy (ICP) was employed to analyze the sodium ion removal capacity and potassium ion exchange capability of the KMAA hydrogel at various initial sodium ion concentrations and pH values.

However, some critical issues remain to be solved and a thorough revision was needed:

1. The “introduction” and “results and discussion” sections of the manuscript can be strengthened and supported with this paper related to the literature and cited (optional for authors): Appl. Organomet. Chem. 37 (2023) e7113.

2. The novelty of this study should be inserted in the text clearly.

3. The advantages and disadvantages of the used adsorbent should be investigated.

4. The mechanism of adsorption should be presented in detail. Cite this ref.: Appl. Organomet. Chem. 36 (2022) e6660.

5. The stability of the adsorbent after the adsorption process should be investigated by XRD.
6. The ion leaching from the adsorbent during the adsorption process should be indicated.

7. The authors should make comparision with literature.

8. The manuscript needs thorough revision to improve the text quality and readability of the work.

Minor editing of English language required

Reviewer 2 Report

The paper is acceptable for publish after several corrections.

The paper mention since the beginning the word KMMA without any other explanation or definition or full name, but it explains the meaning until page 13 in conclusions. It is necessary the full name since the first time it is named.

Did the authors was sure the distilled water have not any level of Na?

Page 4 line 2. The authors mention dry soil and then many times the word soil, and then the preparation; but it is not clear what kind of soil? And where they obtained it. It is necessary mention and define it.

Page 5 line 208, and several times in other lines, for example in line 209 and 216.  It is strange the use of the word shark and sharking instead of stirring, it is better to change it.

Page 7, line 282, it would be good to mention how it was calculated Ki the Langmuir constant or the reference of it; also, the same for 1/n and Kf.

Page 12 fig 6a and 6b. Please clarify. Because it is no clear enough better absorption capacity have better removal efficiency. Maybe write: better capacity absorption in the soil of what? better removal in the solution of what? The same in the other figures of absorption and removal.

Reviewer 3 Report

1、It is suggested to add some experiments on the characterization of hydrogel samples to facilitate the analysis of subsequent series of experimental results.

2、In the experimental methods section, please indicate the clear composition of the solution in 2.41, 2.52 and 2.53 to avoid misunderstanding, and pay attention to the title symbol and similar formatting errors in 2.52.

3、Hydrogel swelling degree if the soil sodium chloride content is high, whether the hydrogel soil will maintain a high water retention rate.

4、Both the left and right sides of the vertical coordinate in Fig. 2 indicate the ion removal rate. It is suggested to add which ion removal rate. The format of the curve in Fig. 2 is inconsistent with that of the curve in the ruler. Please note that the curve format of pH=4 and pH=7 should be modified.

5、The scale format of the pH=4 and pH=7 fitted curves in Fig. 3 is a smooth curve, not a point, please note the same problem in the subsequent figure.

6、Please pay attention to the symbol problems of some cations and ions in 3.5, and the font format should be superscript. Please check similar problems in the whole text and modify them.

7、The horizontal and vertical coordinate format of the picture in Fig. 7 should be unified.

Reviewer 4 Report

Comments

The comments for the manuscript, ‘A novel composite hydrogel material for sodium removal and potassium provision’ are given below,

·         As there are innumerable number of research articles previously published related to hydrogel for metal removal, the word ‘novel’ is not appropriate. Modification of the title is necessary.

·         Line 124 stating, ‘A hydrogel weighing 0.02 g (??) was added to 40 ml of sodium chloride solutions of varying concentrations’. Mention the concentration of NaCl used.

·         In a similar way, NaCl is used for many experiments in the manuscript, but its concentration is not mentioned anywhere. Input the concentration of all the solutions used.

·         For many of the experiments like swelling degree of hydrogel, soil water holding rate, etc, the gel is air-dried and then powdered form of the gel is used. Why didn’t you use the synthesized hydrogel as such? What is the purpose of using dried and powdered form of the gel? Explanation is needed.

·         Several experiments, including swelling and adsorption are theoretically oriented and there are no characterizations related to research oriented, like BET, SEM etc., to increase the quality of the paper.

·         There are no structure confirmation related to the synthesized hydrogel. Include IR and NMR to predict the structure of the hydrogel.

·         Some images of the hydrogel before and after each different experiment is necessary to see any changes in the hydrogel morphology is better for understanding. 

Minor editing of English language required

Round 2

Reviewer 1 Report

Accept

Reviewer 4 Report

The authors have answered my questions, and the manuscript is now acceptable.

Minor editing of English language required